Effects of handling and short-term captivity: a multi-behaviour approach using red sea urchins, Mesocentrotus franciscanus

Bose Aneesh P.H. 1
Zayonc Daniel 2
Avrantinis Nikolaos 2
Ficzycz Natasha 2
Fischer-Rush Jonathan 2
Francis Fiona T. 3
Gray Siobhan 2
Manning Faye 2
Robb Haley 2
Schmidt Coralee 2
Spice Christine 2
Umedaly Aari 2
Warden Jeff 2
Côté Isabelle M. imcote@sfu.ca 3
1 Department of Collective Behaviour, Max Planck Institute for Ornithology , Konstanz , Germany
2 Bamfield Marine Sciences Centre , Bamfield , Canada
3 Department of Biological Sciences, Simon Fraser University , Burnaby , Canada
Taylor Richard
Electronic publication date: 2019 Mar 20
Publication date: 2019
Volume: 7
Electronic Location ID: e6556
Received 2018 Nov 9; Accepted 2019 Feb 2
Copyright: ©2019 Bose et al.
Copyright year: 2019
Copyright holder: Bose et al.
License: This is an open access article distributed under the terms of the Creative Commons Attribution License, which permits unrestricted use, distribution, reproduction and adaptation in any medium and for any purpose provided that it is properly attributed. For attribution, the original author(s), title, publication source (PeerJ) and either DOI or URL of the article must be cited.
License URL: https://creativecommons.org/licenses/by/4.0/

Keywords: Invertebrate, Reintroduction, Relocation, Echinoderm, Stress, Animal welfare

Funding: The authors received no funding for this work.

==============================
Understanding the effects of captivity-induced stress on wild-caught animals after their release back into the wild is critical for the long-term success of relocation and reintroduction programs. To date, most of the research on captivity stress has focused on vertebrates, with far less attention paid to invertebrates. Here, we examine the effect of short-term captivity (i.e., up to four days) on self-righting, aggregation, and predator-escape behaviours in wild-caught red sea urchins, Mesocentrotus franciscanus, after their release back into the wild. Aggregation behaviour, which has been linked to feeding in sea urchins, was not affected by handling or captivity. In contrast, the sea urchins that had been handled and released immediately, as well as those that were handled and held captive, took longer to right themselves and were poorer at fleeing from predators than wild, unhandled sea urchins. These results indicate that handling rather than captivity impaired these behaviours in the short term. The duration of captivity did not influence the sea urchin behaviours examined. Longer-term monitoring is needed to establish what the fitness consequences of these short-term behavioural changes might be. Our study nevertheless highlights the importance of considering a suite of responses when examining the effects of capture and captivity. Our findings, which are based on a locally abundant species, can inform translocation efforts aimed at bolstering populations of ecologically similar but depleted invertebrate species to retain or restore important ecosystem functions.

Introduction

The reintroduction or relocation of wild-caught animals back into the wild can be an effective means for ecological restoration and population management (Van Wieren, 2006; Teixeira et al., 2007). With climate change projections predicting alterations to current ecosystems and species assemblages, the translocation of taxa may become an increasingly important tool for preserving biodiversity and ecosystem functions in the future (Braidwood et al., 2018). However, many relocation programs are associated with unacceptably high rates of mortality among the released animals, and a leading cause of this problem is thought to be stress (Teixeira et al., 2007; Dickens, Delehanty & Romero, 2010). Animals collected for eventual release into the wild experience stress from multiple sources, including their capture, handling, transportation, housing, release, and possible acclimatization to a new environment (Teixeira et al., 2007; Parker et al., 2012). Despite the high priority placed on minimizing the stress associated with any one of these sources, the effects of multiple stressors can be additive and therefore still lead to increased mortality or decreased reproductive success after release (Moberg, 2000).

Most organisms can tolerate acute, brief periods of stress, such as encounters with predators or exposure to short-term adverse environmental conditions (Brown, Gardner & Braithwaite, 2005; Teixeira et al., 2007). The behavioural and physiological changes observed during periods of acute stress are typically considered to be adaptive, as they tend to improve the odds of survival in response to immediate life-threatening situations (Romero & Wingfield, 2015). However, capture and captivity can expose animals to acute and chronic stress (Morgan & Tromborg, 2007) and can impose biological costs; to cope with a chronic stressor, individuals must redirect resources away from normal biological functions, which can influence somatic growth and maintenance (e.g., Lattin et al., 2012), reproduction (e.g., Lombardo & Thorpe, 2009), and sometimes immune response (e.g., Buehler et al., 2008). Captive animals under stress may also exhibit elevated heart rates and high levels of glucocorticoid hormones (e.g., Dickens & Romero, 2009; Fischer, Wright-Lichter & Romero, 2018). Thus, any stress caused during the capture, handling, and captive housing of wild animals can potentially have fitness consequences after their reintroduction to the wild.

How long an animal can be held in captivity while minimizing stress and adverse effects on survival and reproduction upon relocation back to the wild is a pertinent question in conservation biology (Batson et al., 2017). Even short-term captivity, on the order of hours to days, has been shown to cause a significant and detrimental stress response in some animals (e.g., whistling frogs, Litoria ewingi, Coddington & Cree, 1995; loggerhead sea turtles, Caretta caretta, Gregory et al., 1996). To date, studies investigating the effects of stress on released animals have largely focused on vertebrates—a pattern often seen in other branches of science (Andrews, 2011) and in conservation mandates—leaving a considerable gap in knowledge pertaining to invertebrates.

Here, we examine whether time spent in captivity affects the behaviour of a wild-caught invertebrate upon its release back into the wild. Using the red sea urchin, Mesocentrotus franciscanus, as a model species, we investigated how short-term captivity of up to four days affects three behaviours potentially related to fitness. Specifically, we examined their propensity to aggregate and two anti-predation behaviours: self-righting and predator-escape responses. We considered the propensity to aggregate as a potential index of feeding behaviour since several species of sea urchins, including red sea urchins, aggregate in areas of patchy food sources such as around drift algae (Lees, 1970; Russo, 1979; Vadas et al., 1986). Righting refers to the ability of sea urchins to return to a normal ‘oral surface down’ position from an inverted position, which exposes the less-defended oral surface (Polls & Gonor, 1975). Finally, a common escape response of sea urchins in the presence of predatory sea stars is to rapidly move away (Vadas Sr & Elner, 2003; Urriago, Himmelman & Gaymer, 2011). We viewed any impairment of the righting or fleeing response as a potential increase in the vulnerability to predation of released sea urchins. We predicted that if captivity induces chronic stress, then the ability of sea urchins to perform these three key behaviours after release back into the wild should diminish as length of captivity (i.e., duration of stress) increases. Importantly, we also examined the individual effects of handling and tagging the animals, to distinguish them from the effects of captivity.

Materials & Methods

Ethics statement

All procedures used in this study adhered to the ethical considerations and guidelines set by the Canadian Council on Animal Care (CCAC) and the animal ethics committee at the Bamfield Marine Sciences Centre (AUP UP16-SP-SCD-01 and UP16-SP-SCD-02).

Urchin collection and captivity conditions

A total of 198 red sea urchins were collected between 11 May and 17 May 2016 from Aguilar Point (48°50.381′N, 125°8.477′S) near the Bamfield Marine Sciences Centre (BMSC) in British Columbia, Canada. Sea urchins were collected haphazardly by five pairs of scuba divers from depths ranging from 6 to 9 m. We divided the collected sea urchins into five experimental groups, which were each held in captivity for differing lengths of time before release back into the wild: zero-day (n = 38, ‘handling controls’ described below), one-day (n = 40), two-day (n = 40), three-day (n = 40), and four-day (n = 40) captivity periods. Each group was collected on a different day to allow for a staggered release of the sea urchins back into the wild and to allow time for follow-up behavioural testing post-release.

We transported all animals to be held in captivity (i.e., one- to four-day groups) in seawater-filled coolers, and housed them at the BMSC in large, aerated water tables (170 × 70 × 20 cm) fitted with a flow-through sea water system (water temperature: 11  ± 0.5 °C). We placed each experimental group in a separate water table. These groups were not fed for the duration of their captivity, meaning that captivity co-occurred with food deprivation in our experiments. While we did not parse out the specific consequences of starvation here, we did not expect that such short-term food deprivations would dramatically impair urchin behaviour because sea urchins are highly resilient to starvation. For example, both red sea urchins and purple sea urchins, S. purpuratus, have been shown to survive up to five months of starvation, although with growth and reproductive consequences (Holland, Giese & Phillips, 1967; Bureau, 1996). On their day of release, we measured the test diameter of each sea urchin to the nearest mm using calipers (mean test diameter  ± sd = 59.6  ± 11.4 mm, range = 23–95 mm), and gave each sea urchin two identically-coded silicon tags (∼0.5 × 0.5 cm, <200 mg in air, placed halfway down each of two separate spines; Fig. 1), allowing us to identify individuals post-release in the wild. Sea urchins in the 0-day ‘handling control’ group served as a procedural control, allowing us to separate the effects of handling from those of captivity. These control animals were not held in captivity; instead, they were held in water-filled coolers at the surface (for ∼1 h) while being tagged and then immediately released.

Figure 1 Tagged red sea urchin released in the wild.

Every urchin was marked with individually numbered silicon tags. Photo credit: IM Côté.

Sea urchin release and behaviour assessment

Release of the experimental groups of tagged sea urchins occurred at the same site as they were collected, at five anchored buoys placed 10 m apart at constant depth (∼6 m below chart datum). When the designated captivity period for an experimental group had elapsed, seven to eight sea urchins were released using SCUBA at the base of each buoy (summing to 38–40 sea urchins per experimental group). Immediately following their release, we tested the ability of the sea urchins to right themselves. To do this, we set each sea urchin with its aboral side down on a relatively flat rock, and righting speed was measured as the time needed to return to their normal position with their oral surface against the substrata. In a pilot study, we found that most sea urchins could right themselves within 120 s. We therefore set a 180-s limit for this assay and recorded if any sea urchins were unable to right themselves in this time.

Twenty-four hours post-release, we relocated as many of the tagged sea urchins as possible that we had released on the previous day by thoroughly searching the area within a 5-m radius of each buoy, and more haphazardly searching the area beyond. Upon relocating a tagged sea urchin, we assessed its propensity to aggregate with other sea urchins by counting the number of other red sea urchins within a 1-m radius of the tagged animal. This method is a spot estimate of density and we use it here as a proxy for local sea urchin aggregation (henceforth called ‘aggregation score’). Previous studies have similarly investigated sea urchin aggregation at this spatial scale, i.e., by using 1 m2 quadrats (e.g., Russo, 1979). Righting times were measured again, as described above. Finally, we also measured each sea urchin’s predator escape response by placing the tagged sea urchin on a flat surface along a measuring tape and gently touching, for approximately 1–2 s, one side of the sea urchin with an arm of a predatory sunflower sea star, Pycnopodia helianthoides (Mauzey, Birkeland & Dayton, 1968). Such an action generates a chemosensory-induced flight response in sea urchins that are preyed upon by P. helianthoides (Moitoza & Phillips, 1979). Predatory sea stars were collected earlier from the same location and were 50–80 mm in diameter. The distance travelled by the sea urchin along the tape was measured for 30 s after contact with the sea star.

Wild unhandled sea urchin controls

On each day that we tested the behaviour of our released (and relocated) treatment sea urchins (i.e., the zero-day ‘handling controls’ as well as the one-day, two-day, three-day, and four-day captives), we also conducted behavioural tests on wild, unhandled, and untagged sea urchins found nearby. These sea urchins represented another control group, henceforth called ‘wild control’, which allowed us to examine the effects of handling itself. Here, handling refers to the actions of ascending with the animal to the surface for tagging and then descending again for release. We selected every third untagged sea urchin (within the size range of the experimental sea urchins) that we encountered while roving in the immediate vicinity of the anchored buoys. The selected untagged sea urchins were also at least 1 m away from any other untagged sea urchin that we had previously assessed. We measured the righting times, aggregation scores, and predator escape responses for these wild control sea urchins (6–19 sea urchins tested per day for a total of 64).

Assessing the effects of tagging

To examine the effects of tagging, we collected an additional 20 red sea urchins from Aguilar Point, which were kept for five days in individual tanks at BMSC fitted with flow-through sea water and held at 11 ± 0.5 °C. Ten of the sea urchins were tagged, as described above, and their righting response was tested (as above, but with no time limit). The tags were then removed and the righting time measured again. The order of testing was reversed for the other 10 individuals; their righting times were measured without tags and then again after tagging. This series of behavioural tests was conducted on these individuals after one and five days of captivity, with the sea urchins bearing tags for the duration in between their tests.

Statistical analyses

All analyses were performed in R version 3.4.4 (R Core Team, 2018). Note that because of time constraints while diving and the fact that not all tagged sea urchins could be relocated 24 h post-release, sample sizes vary slightly across analyses.

We first tested whether the proportions of sea urchins that succeeded in righting themselves within 180 s varied across experimental groups (i.e., ‘wild control’, zero-day ‘handling control’, one-day, two-day, three-day, and four-day captive) using chi-squared tests for equality of proportions (with Yate’s continuity corrections). We then examined whether sea urchin righting times differed among experimental groups (for those sea urchins that righted themselves within 180 s). To do this, we fit two linear mixed-effects models (LMMs, ‘nlme’ R package, version 3.1–131.1, Pinheiro et al., 2018), one to the righting times measured immediately upon release and another to the righting times measured 24 h post-release. The responses were log-transformed to improve the normality and homoskedasticity of the model residuals based on Shapiro–Wilk tests and examination of standardized residuals versus fitted values plots. We included treatment group and test diameter as fixed effects and ‘diver buddy-pair ID’ as a random intercept to account for variation among divers in their handling of the animals and among sites where the data were collected. We then examined the following pair-wise contrasts using Dunnett’s comparisons: (1) wild controls versus each of the remaining experimental groups, and (2) handling controls versus each of the remaining experimental groups (using ‘glht’ function in ‘multcomp’ R package, version 1.4-8, Hothorn, Bretz & Westfall, 2008). We then tested whether urchin aggregation scores and predator escape responses differed among our experimental groups. To do this, we similarly fit LMMs to these data using the same fixed and random effects structures as described previously and followed these up by examining pair-wise Dunnett’s contrasts.

Lastly, we tested whether the silicon tags used for individual identification in the wild influenced sea urchin behaviour, in particular their righting times. We fit a LMM to the sea urchin righting times measured in the lab (log-transformed). We included the day of behavioural testing (i.e., day 1 vs. day 5), sea urchin test diameter, and tag status (i.e., tagged vs. untagged) as fixed effects, as well as ‘urchin ID’ as a random intercept.

Results

Effects of handling and captivity on sea urchin behaviour

Overall, 12% of all sea urchins tested (31 out of 262 sea urchins) did not right themselves within 180 s on the first test (i.e., immediately upon release of captive sea urchins). There was initial variation in righting success among experimental group (χ52=13.9, P = 0.016). This effect appeared to be largely driven by the high rate of righting failure of sea urchins in the handling control group (26%, or 10 out of 38 sea urchins, vs 10% or 16 out of 160 sea urchins across other groups). When the handling control group was omitted, we no longer detected a difference between groups in righting success (χ42=6.1, P = 0.18). On the second test (24-h post-release), only 8% of sea urchins did not right successfully (13 out of 163 sea urchins), and there was no difference among experimental groups in the proportions of sea urchins that successfully righted themselves (χ52=8.7, P = 0.12).

Untagged, unhandled sea urchins (i.e., wild controls) were consistently faster at righting themselves than each group of handled and captive sea urchins tested immediately upon their release (LMM, all contrasts were at least: est. ± se = 0.19 ± 0.04, z = 4.4, P < 0.001) (Fig. 2A). Sea urchins that served as handling controls, i.e., tagged and released immediately, had similar righting times to all captive-held sea urchin groups upon release (all contrasts were at least: est. ± se = 0.09 ± 0.06, z = 1.6, P = 0.34) (Fig. 2A). After 24 h, wild controls were still significantly faster at righting themselves than the handling control group (est. ± se = −0.44 ± 0.14, z =  − 3.1, P = 0.01), the 1-day (est. ± se = 0.48 ± 0.13, z = 3.7, P = 0.001), 2-day (est. ± se = 0.34 ± 0.13, z = 2.6, P = 0.05) and 4-day captive sea urchins (est. ± se = 0.36 ± 0.13, z = 2.8, P = 0.03), but not the 3-day captive sea urchins (est. ± se = 0.23 ± 0.13, z = 1.8, P = 0.26) (Fig. 2B). The righting time of handling control sea urchins did not differ from those of any of the captive-held sea urchin groups after 24 h (all contrasts were at least: est. ± se = −0.21 ± 0.16, z =  − 1.3, P = 0.52) (Fig. 2B).

Figure 2 Sea urchin righting times measured (A) immediately after release into the wild and (B) 24 h post-release.

Capital letters denote the results of comparisons between ‘wild’ controls as the reference group and all other experimental groups; lowercase letters denote the results of comparisons between ‘handling’ controls as the reference group and all captive experimental groups. Within a comparison set, means with similar letters are not significantly different from the reference group. Bars represent means ± standard error (in s). Sample sizes per group are given above each bar in parentheses.

Mean aggregation scores for each sea urchin group varied from 13 to 17.4 individuals within a 1-m radius of a focal sea urchin. There were no detectable differences in sea urchin aggregation scores between any of the experimental groups and the wild controls (LMM: all contrasts were at least: est. ± se = 2.32 ± 1.26, z = 1.8, P = 0.25) or the handling controls (all contrasts were at least: est. ± se = 3.38 ± 1.41, z = 2.4, P = 0.06) (Fig. 3).

Figure 3 Sea urchin aggregation scores.

Capital letters denote the results of comparisons between ‘wild’ controls as the reference group and all other experimental groups; lowercase letters denote the results of comparisons between ‘handling’ controls as the reference group and all captive experimental groups. Within a comparison set, means with similar letters are not significantly different from the reference group. Bars represent means ± standard error. Sample sizes per group are given above each bar in parentheses. Aggregation scores were obtained by counting the number of sea urchins within a one-meter radius of focal urchins on each sampling day.

When induced to flee by contact from a predatory sea star, wild control sea urchins moved farther in 30 s than the 1-day (LMM: est. ± se = 5.39 ± 1.98, z = 2.7, P = 0.03) and the 2-day captive sea urchins (est. ± se = 5.53 ± 2.00, z = 2.8, P = 0.03), but not the 3-day (est. ± se = 4.10 ± 1.96, z = 2.1, P = 0.15) or 4-day captive sea urchins (est. ± se = 1.50 ± 1.99, z = 0.75, P = 0.93), or the handling control sea urchins (est. ± se = 5.20 ± 2.14, z = 2.4, P = 0.07) (Fig. 4). Sea urchins that served as handling controls had a similar predator escape response to the other experimental groups (all contrasts were at least: est. ± se = 3.70 ± 2.22, z = 1.7, P = 0.30).

Figure 4 Distance travelled by sea urchins in 30 s after being induced to flee by contact with a predatory sea star.

Capital letters denote the results of comparisons between ‘wild’ controls as the reference group and all other experimental groups; lowercase letters denote the results of comparisons between ‘handling’ controls as the reference group and all captive experimental groups. Within a comparison set, means with similar letters are not significantly different from the reference group. Bars represent means ± standard error (in cm). Sample sizes per group are given above each bar in parentheses.

Effect of tagging on sea urchin righting times

Red sea urchins in captivity took an average (±sd) of 157 s (±135 s) to right themselves in the laboratory. Tagging did not affect the sea urchins’ ability to right themselves after one day or five days of captivity (tagged vs untagged, LMM: est. ± se = 0.014 ± 0.076, t58 = 0.19, P = 0.85). The length of captivity (1-day vs 5-days) also did not affect righting times (est. ± se = 0.032 ± 0.076, t58 = 0.42, P = 0.67). Larger sea urchins righted themselves faster than smaller sea urchins (est. ± se = −0.009 ± 0.003, t18 = −2.96, P = 0.008).

Discussion

Using a series of behavioural tests conducted in the field and in the laboratory, we examined the effects of handling, tagging, and captivity on the behaviour of wild-caught and released red sea urchins. We show that handling the animals impaired, at least in the short term, two of the three behaviours examined: self-righting and predator escape speed. Tagging had no effect on the righting behaviour of captive sea urchins, and the length of captivity (i.e., the length of the putative stress period) did not have a detectable effect on sea urchin behaviour post-release, for periods of up to four days of captivity. The detectable effect of handling in combination with a lack of effect of duration of captivity suggests that starvation—a potentially confounding factor in our experiments—played little or no role in altering sea urchin behaviour. Longer-term monitoring of survival and reproduction of released animals is needed to establish whether the short-term behavioural impairments observed have fitness consequences.

Overall, the vast majority (96.9%) of sea urchins in our study moved away from the predatory sea star stimulus (i.e., they exhibited a flight response). This suggests that the primary anti-predator behaviour among our sampled population of sea urchins was to flee as opposed to stay and combat the sea star with their spines (an alternative anti-predator defence behaviour of M. franciscanus suggested by Moitoza & Phillips, 1979). We show that wild, unhandled red sea urchins were faster at moving away from the predator stimulus than sea urchins that were first handled, including those that were handled and held in captivity. Handled and captive-held sea urchins were also slower at righting themselves, and these effects persisted for at least one day after release. Thus, sea urchins that experienced capture and handling might be more susceptible to predation than their unhandled counterparts after release. Impaired anti-predator behaviour has been noted in captured animals that are subsequently released (e.g., Raby et al., 2014). For example, wild reef fish that were caught and taken out of water took longer to reach the reef and to seek shelter after being released than wild fish exposed to less stressful conditions (Raby et al., 2018). Caribbean spiny lobster Panulirus argus captured and released by divers abandoned previously occupied shelters, which might have contributed to their decreased survival when predators were present (Parsons & Eggleston, 2006). Similarly, translocated cane toads Rhinella marina selected shallower shelters and spent more time visible by day than resident toads, which might have made the former easier for predators to locate (Pettit, Greenlees & Shine, 2017). At least in vertebrates, post-release behavioural deficiencies are usually ascribed to physiological, locomotory and/or cognitive impairments associated with the stress of capture (Cooke et al., 2013; Raby et al., 2014).

Animals that are captured to be released at a later date are sometimes tagged to allow subsequent recognition and monitoring. Tagging can cause a number of behavioural alterations, ranging from changes in foraging, parental care, movement and habitat choice (Walker et al., 2012; Jepsen et al., 2015), but disentangling the effects of tagging from those of capture and captivity can be challenging and is seldom done (Jepsen et al., 2015). We were able to show that tagging sea urchins had a non-detectable effect on their righting times in the laboratory. This result lends confidence that the slower righting times of captured and released sea urchins were not due to tagging but to handling, which in our case meant bringing the animals to the surface from depths of 6 - 9 m for ∼1 h and releasing them back to the capture point. The tags we used likely did not have a behavioural effect on red sea urchins because they were non-invasive (see also Dumont, Himmelman & Russell, 2006, who used beaded monofilament on the sea urchin Strongylocentrotus droebachiensis). In contrast, more invasive tags, such as internal tags or branding, can have marked effects on the behaviour, growth and survival of sea urchins (Lauzon-Guay & Scheibling, 2008) and other aquatic invertebrates (e.g., Wilson et al., 2011; Martinez, Byrne & Coleman, 2013).

We had predicted that any behavioural impairments observed in released sea urchins might increase as the length of captivity, or stress period, increased. This was not the case. The duration of captivity had no detectable effect on sea urchin righting time, either upon release or 24 h later. In contrast, the predator escape response of released sea urchins appeared to improve after two days in captivity, such that sea urchins that had been in captivity for three or four days moved away from the predator stimulus at similar speeds to unhandled wild sea urchins. Our behavioural results therefore suggest a possible reversal of the stress state of sea urchins held in captivity for more than 48 h. In many vertebrates, the initial hours or days of captivity can result in weight loss, increases in stress hormone (i.e., cortisol, corticosterone) concentrations, and disruption of the negative feedbacks controlling stress hormone production (e.g., Davidson, 1984; Coddington & Cree, 1995; Davidson et al., 1997; Dickens, Earle & Romero, 2009; Adams et al., 2010). However, these changes can reverse after a few days of captivity as seen in some species of birds, for example after three days in North Island saddlebacks, Philesturnus rufusater (Adams et al., 2010) and nine days in chukar, Alectoris chukar (Dickens, Earle & Romero, 2009), indicating acclimation to captive conditions. To our knowledge, a similar pattern has not been previously noted in invertebrates. Future studies should investigate the effects of longer periods of captivity, and assay a suite of stress responses and behaviours involved in foraging, reproduction, and defense against multiple predator types, to determine the generality and trend shape (i.e., stepwise or gradual) of potential acclimation to captivity in invertebrates.

While handling affected the self-righting and predator escape responses of red sea urchins, neither handling nor captivity altered their propensity to aggregate with conspecifics. Aggregation in this species and in other sea urchins has been shown to be closely associated with food abundance (Russo, 1979; Vadas et al., 1986). We had therefore hypothesised that stress from handling and captivity might diminish foraging behaviour, as it sometimes does in vertebrates (Morgan & Tromborg, 2007), leading to reduced aggregation. It is possible that no effect was observed because the link between sea urchin foraging and aggregation is geographically or seasonally variable (e.g., Lauzon-Guay & Scheibling, 2007) or strongly modified by biotic factors, such as the presence of predators or the dispersion of food sources (e.g., Bernstein, Schroeter & Mann, 1983). The high density of red sea urchins and the dispersed distribution of kelp (APH Bose, D Zayonc, N Avrantinis, N Ficzycz, J Fischer-Rush, FT Francis, S Gray, F Manning, H Robb, C Schmidt, C Spice, A Umedaly, J Warden, IM Côté, pers. obs., 2016) at our study site might have reduced the incentive for sea urchins to aggregate.

Overall, our results indicate that the duration of (short-term) captivity does not cause detrimental effects on the post-release behaviour of wild-caught red sea urchins, at least for those behaviours we tested. However, handling alters, at least in the short term, behaviours that could have fitness consequences, although longer-term studies on behaviour, reproduction and survival will be needed to verify this. Two of the three behaviours examined were affected, highlighting the usefulness of considering a suite of responses. While there might be limited interest, other than for scientific study, in the capture and release of a species as abundant as M. franciscanus, our findings could inform conservation efforts aimed at ecologically similar but depleted species. Examples include the Caribbean long-spined sea urchin, Diadema antillarum, decimated by disease in the 1980s (Lessios, 1988), and the European purple sea urchin, Paracentrotus lividus, which has collapsed in parts of the Mediterranean Sea due to disease linked to warming temperatures (Girard et al., 2012; Yeruham et al., 2015). Translocation of individuals of such species could bolster local populations or concentrate remaining individuals in more favourable habitats, restoring the ecosystem functions (e.g., herbivory) provided by these ecologically important species.

Supplemental Information

Supplemental Information 1 Urchin field data

Click here for additional data file.

Supplemental Information 2 Urchin tag validation lab data

Click here for additional data file.

We thank the Huu-ay-aht First Nations for their permission to sample sea urchins from their territory, and the Bamfield Marine Sciences Centre (BMSC) and its staff for their time, resources, use of animal-holding facilities, and hospitality. Thanks to Steve Johnson and Janice Pierce for providing boat tending and surface safety support during diving sessions.

Additional Information and Declarations

Competing Interests

Author Contributions

Field Study Permissions

Data Availability

The authors declare there are no competing interests.

Aneesh P.H. Bose and Daniel Zayonc conceived and designed the experiments, performed the experiments, analyzed the data, prepared figures and/or tables, authored or reviewed drafts of the paper, approved the final draft.

Nikolaos Avrantinis, Natasha Ficzycz, Jonathan Fischer-Rush, Faye Manning, Haley Robb, Coralee Schmidt, Christine Spice, Aari Umedaly, Jeff Warden conceived and designed the experiments, performed the experiments, authored or reviewed drafts of the paper, approved the final draft.

Fiona T. Francis, Siobhan Gray and Isabelle M. Côté conceived and designed the experiments, performed the experiments, contributed reagents/materials/analysis tools, authored or reviewed drafts of the paper, approved the final draft.

The following information was supplied relating to field study approvals (i.e., approving body and any reference numbers):

All procedures used in this study adhered to the ethical considerations and guidelines set by the Canadian Council on Animal Care (CCAC) and the Animal Ethics Committee at the Bamfield Marine Sciences Centre (AUP UP16-SP-SCD-01 and UP16-SP-SCD-02).

The following information was supplied regarding data availability:

Cote, Isabelle (2019): Urchin data—for repository.csv. figshare. Dataset. https://doi.org/10.6084/m9.figshare.7302374.v1.

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
