# Peer review of "Effects of handling and short-term captivity: a multi-behaviour approach using red sea urchins, Mesocentrotus franciscanus"

_PeerJ, doi:10.7717/peerj.6556_

## Round 0.1 · original submission · Minor Revisions

Both reviewers liked your MS and have a made a number of constructive suggestions. Please address these in your revision.

·

Basic reporting

This was a nice clear manuscript, well written and prepared. I enjoyed reading. I apologise to the authors for my delay in getting the review in. This is in no way the fault of the journal

Experimental design

Sound. I do think the story is not as clear as the authors propose, since the animals were both starved and held in captivity. This needs to be clarified

Validity of the findings

Greater context of translocation

Additional comments

General Comments

This paper reports three neat small experiments to test the effects of handling and captivity on short-term organism performance. Usefully, the authors use a tractable invertebrate species to achieve better ecological representation rather than focussing on the cute and fluffy. The authors have missed a context opportunity – translocation of taxa has been suggested as strategy to reduce the effect of climate change on ecosystems (Braidwood et al. 2018), and usefully, the authors have produced data that can inform this debate. The work is competently done and well presented. I do have one small issue that can affect interpretation. The animals held captive were also starved, so in reality, the experiments tested for the effect of captive animals held and handled with no food, rather than captive animals with access to nutritional resources. Some commentary is needed.

Detailed comments
• Line 39. The link to fitness is tenuous at best. Downplay
• Line 52 and other places following. Please avoid starting a sentence with “however” – except under the circumstance… ‘However much John ate, he could not stop feeling hungry’
• Line 60. Rephrase. Evolution has no foresight, hence animals cannot be selected to do stuff
• Line 61. Adverse environmental conditions will also be sources of stress
• Lines 90-97. The link between the behaviours measured and what can be inferred from them is not clear. This section needs strengthening
• Lines 117 – 127. As indicated above, the animals are both starved and in captivity, so you are not just measuring captivity
• Line 123. Greater discussion of the tags or point to a reference.
• Line 138 and elsewhere – animals live on or in substrata (singl. Substratum), enzymes act on a substrate.
• Line 138. Why 180 s?
• Line 146. This method of estimating aggregation has little support in the literature (see Krebs 1999, for an excellent discussion). With no comparable data for the locality, what is actually being measured here is a spot estimate of density, rather than some estimate of frequency distributions of inter-animal distances. Clarify that is a proxy for aggregation, but please don’t refer to it as an estimate of aggregation for a local population.
• Line 214 and following: It is illogical and silly to give (pseudo)exact probability values for frequentist tests of hypotheses. This is because the logic of hypothesis testing only requires that the probability is less than a given alpha. The silliness is because the P value doesn’t actually tell you anything; in frequentist statistics the P value gives the probability of getting that result if the null hypothesis is correct and the experiment were to be repeated many times. It does not give the probability of the null hypothesis being correct. Please change throughout. Refer to the American Statistical Association statement at http://amstat.tandfonline.com/doi/abs/10.1080/00031305.2016.1154108#.Vt2XIOaE2MN
• Line 263. To properly measure fitness, you need to measure reproduction (hard, but not impossible in sea urchins) see Don Levitan’s papers (e.g. 2002; 1992)
• The figures were clear, thank you.



Braidwood, David W., Mark A. Taggart, Melanie Smith, and Roxane Andersen. 2018. "Translocations, conservation, and climate change: use of restoration sites as protorefuges and protorefugia." Restoration Ecology 26 (1):20-8. doi: doi:10.1111/rec.12642.
Krebs, C.J. 1999. Ecological Methodology. 2nd ed. Menlo Park: Addison-Wesley.
Levitan, D. R. 2002. "Density-dependent selection on gamete traits in three congeneric sea urchins." Ecology 83 (2):464-79.
Levitan, D.R., M.A. Sewell, and F.S. Chia. 1992. "How distribution and abundance influence fertilization success in the sea urchin Strongylocentrotus franciscanus." Ecology 73:248-54.

·

Basic reporting

The article is well-written and polished. It was easy to read and flowed well. I did not see that the raw data was shared, however.

Experimental design

Research questions were well-defined and the methods were well-designed and clearly described.

Validity of the findings

Data appear to be robust, statistically sound, and controlled. Conclusions are clear and well-supported by their results.

Additional comments

Overall this study was well-designed, the paper was well-written and very easy to follow. This study is relevant and acceptance and publication is recommended with just a few minor revisions.

Specific comments:
Line 325 I might suggest that this sentence be revised since it is impossible to test all possible post-release behaviors.

Line 333 I believe the common name for D. antillarum is long-spined sea urchin. In addition this sentence appears to be an incomplete sentence.

The results for the distance travelled by sea urchins in 30 s after being induced to flee by contact with a predatory sea star (Fig. 3) were unexpected to me. The authors say that the urchins held in captivity appeared to recover from the handling event and “improved gradually after two days in captivity” (lines 302-304). Could this particular upward trend in the distance traveled be tested statistically? Is the slope different from zero? This was a very interesting result that could be highlighted further perhaps with a regression.

Then they suggest that there is a “possible reversal of the stress state”, however have other studies investigated this predator escape response of urchins held during longer periods. Do the authors think that this particular behavior needs to be investigated further? Can they suggest other methodologies in which researchers could test fleeing when in contact with a predator? They could include a few sentences on future research ideas, such as flight response from other predators—sheephead for instance may be more challenging to test in the field but perhaps this could be done at least for lobsters in the field.

Could urchins forget how to respond to a predator if held for a very long time? Further, could the captivity itself be a stressful experience if longer than the length of time examined in this study? At the end of this paragraph there are a few references pertaining to bird species, but could the authors include relevant references of studies of marine invertebrates. I know there has been some research on anti-predator responses of snails from water-borne stimuli or cues (for example: Alexander and Covich 1991 https://doi.org/10.2307/1542339). Perhaps a marine invertebrate example may be a bit more relevant than bird references—since the predator cues and captive environment are in the same medium as urchins (water rather than air). If you include the bird references, you may want to say these are birds, in case there are any readers unfamiliar with the common names saddlebacks and chukars.

What other behavioral responses could be tested in future studies?

---

## Round 0.2 · accepted · Accept

Thank you for addressing the reviewers' comments.

#